# Localization Performance Analysis and Algorithm Design of Reconfigurable Intelligent Surface-Assisted D2D Systems

**DOI:** 10.3390/s24113694

**Published:** 2024-06-06

**Authors:** Mengke Wang, Tiejun Lv, Pingmu Huang, Zhipeng Lin

**Affiliations:** 1School of Information and Communication Engineering, Beijing University of Posts and Telecommunications (BUPT), Beijing 100876, China; wmk@bupt.edu.cn; 2School of Artificial Intelligence, Beijing University of Posts and Telecommunications (BUPT), Beijing 100876, China; pmhuang@bupt.edu.cn; 3Key Laboratory of Dynamic Cognitive System of Electromagnetic Spectrum Space, College of Electronic and Information Engineering, Nanjing University of Aeronautics and Astronautics (NUAA), Nanjing 211106, China; linlzp@nuaa.edu.cn

**Keywords:** reconfigurable intelligent surfaces, device-to-device (D2D), millimeter-wave, positioning error bound, beamforming design

## Abstract

The research on high-precision and all-scenario localization using the millimeter-wave (mmWave) band is of great urgency. Due to the characteristics of mmWave, blockages make the localization task more complex. This paper proposes a cooperative localization system among user equipment (UEs) assisted by reconfigurable intelligent surfaces (RISs), which considers device-to-device (D2D) communication. RISs are used as anchor points, and position estimation is achieved through signal exchanges between UEs. Firstly, we establish a localization model based on this system and derive the UEs’ positioning error bound (PEB) as a performance metric. Then, a UE-RIS joint beamforming design is proposed to optimize channel state information (CSI) with the objective of achieving the minimum PEB. Finally, simulation analysis demonstrates the advantages of the proposed scheme over RIS-assisted base station positioning, achieving centimeter-level accuracy with a 10 dBm lower transmission power.

## 1. Introduction

In next-generation wireless networks, achieving high-precision localization in various scenarios remains a focus of further research. Current studies primarily rely on the accurate estimation of channel parameters, such as time of arrival (TOA), direction of arrival (DoA), or received signal strength indicator (RSSI), to enable localization by establishing communication links between transmitters and receivers. Among these, DOA-based methods such as multiple signal classification (MUSIC) [1] and estimation of signal parameters using rotational invariance techniques (ESPRIT) [2] algorithms, which utilize uniform linear arrays (ULA) for parameter estimation, have many applications. The positioning model based on RSSI has also been applied in recent research [3,4]. The RSSI value of radio waves during propagation is a function of the propagation distance, and this relationship can be used to estimate the distance. Utilizing RSSI based on pilot signal power for distance estimation at close range is a good choice, especially when there are no obstacles blocking the signal.

However, practical scenarios often involve obstacles between base stations (BSs) and user equipment (UEs) or mobile stations (MSs), resulting in non-line-of-sight (NLOS) situations. This becomes particularly prominent at millimeter-wave (mmWave) or higher frequencies, where signal penetration is weaker. Fortunately, with the advancements in metamaterials and radio frequency technology, reconfigurable intelligent surfaces (RISs) offer a promising solution to address NLOS localization [5]. Utilizing RISs for localization in mmWave scenarios has become a pivotal area of recent research. This is supported by studies such as [6,7,8], which demonstrate the capability of achieving centimeter-level positioning accuracy, even under conditions of obstructed links.

RIS is a revolutionary technology that intelligently reconfigures the wireless propagation environment by using a large number of low-cost passive reflecting elements integrated on a plane. It significantly enhances the performance of wireless communication networks [9]. RIS with multiple reflective elements provides higher beamforming (BF) gain and angular resolution, which is unmatched by ordinary scattering points. Despite the tremendous potential of RISs, from a communication design perspective, RIS faces new challenges in effectively integrating into wireless networks, such as reflection optimization, channel estimation, and deployment [9]. In recent years, researchers have made lots of meaningful work on RIS-aided communication positioning. The authors in [10] analyzed the impact of RIS-reflected paths on the accuracy of position estimation, but it only considered 2D scenarios. Other authors in [11,12] investigated the Cramér–Rao lower bound for absolute position estimation in 3D scenarios assisted by RIS, and optimized RIS-reflected BF to enhance localization performance. There are also considerations of cooperative localization methods. In [13,14], adding device-to-device (D2D) links between MSs was proposed to assist localization and the authors demonstrated that cooperative communication between MSs can provide additional localization gains through numerical results. However, the studies above are all based on the localization method with the BS as the necessary anchor point. Since RISs’ positions are known, it should be feasible to use RISs as anchor points for localization. In fact, the idea has been explored in [15,16]. The authors in [15] involved a single UE signal which is reflected back to itself through adjusting the RIS elements. After eliminating multipath effects, UE achieved localization itself. Ref. [16] introduced a D2D link based on [15], but its core is the same as [15]. In addition, the reflection design of the RIS is also a crucial aspect of positioning. Both [17,18] have conducted reflection BF designs by solely considering the RIS and utilizing discrete phase shifts. These studies have successfully demonstrated the effectiveness of such discrete phase shift designs. Hence, adopting a discrete design approach for localization is also a viable option.

Motivated by the above, this paper newly proposes a cooperative localization system among UEs with RISs as anchor points and conducts a performance analysis on this system. We consider both positioning relying solely on RIS and cooperative positioning among UEs. The proposed scheme differs significantly from previous papers in terms of details and introduces novel aspects in problem optimization. The main contributions of this paper are summarized as follows:We present a cooperative localization system utilizing RISs for millimeter-wave (mmWave) multi-input multi-output (MIMO) systems. In this system, UEs engage in D2D communication via RISs to facilitate localization, with the RISs serving as anchors without direct communication with BS or any access points. Leveraging RSSI and DOA methods such as ESPRIT, we extract parameters like angle and distance to establish geometric equations. The positions of UEs are then estimated by solving these equations. This proposed localization scheme not only significantly reduces the positioning pressure of the BS but also offers greater localization flexibility for UEs. Furthermore, it can serve as a supplementary means to enhance accuracy in BS-based UE localization.To evaluate the proposed system, we utilize the positioning error bound (PEB) of UEs as a performance metric. To accomplish this, we first calculate the Fisher information matrix (FIM) for the channel parameters. We then derive the transformation matrix between the FIMs of the channel parameters and the UEs’ positions. This allows us to determine the PEB for the UEs. With the aim of reaching the optimal performance of proposed system, we set the PEB as our objective. Since the received signal is significantly influenced by the channel state information (CSI), we innovatively propose a joint BF design of the transmitter UE’s BF and reflecting BF at RIS based on an alternating optimization algorithm. We illustrate our objective function and detail the algorithm steps of the proposed BF design to solve the function.The simulation results illustrate that, with the proposed BF design, our localization scheme achieves markedly superior positioning performance compared to scenarios without BF design or with random BF design. Moreover, by opting for smaller RIS sizes and an optimal number of quantization resolutions, our proposed scheme achieves positioning accuracy of $10^{-2}$ meters under the condition that signal-to-noise ratio (SNR) is 30 dB. This level of accuracy typically necessitates higher power consumption or larger RIS sizes in BS-based user localization systems.

The remainder of this paper is organized as follows. Section 2 introduces the system model, including the channel model, receiver model, and geometry relationship. Preliminary localization is presented, and the PEB is derived in Section 3. Section 4 demonstrates the optimization objective and proposes our algorithm. Section 5 shows the simulation analysis, including the impact of the antenna array size, the number of carriers, different quantization resolutions, and physical distance on PEB. There, we also conduct a comparative experiment. Conclusions are presented in Section 6.

Notations: Vectors are shown by bold lower-case letters and matrices by bold upper-case ones. $j=\sqrt{-1}$. E[·] denotes the expectation operator. ℜ{·} represents the real part of a complex variable. All vectors are column vectors by default. Transpose and Hermitian operations are denoted by ${\left(\cdot \right)}^{T}$ and ${\left(\cdot \right)}^{H}$, respectively. The Hadamard product is indicated by ⊙.

## 2. System Model

We consider a 2D localization scenario between UEs with two RISs assistance and no BS or access point, as presented in Figure 1, which can be called cooperative localization among UEs with RISs as anchors. It consists of RISs with *L* reflecting antenna elements in each ULA and two ULA users UE1 and UE2 with $N_{u}$ antennas each. More users are feasible, but we aim to simplify the model and focus on the analysis of positioning performance. Two or more RIS units are necessary because we need a sufficient number of anchor points to construct the spatial topology of the devices. This conclusion is based on geometric intuition, and no formal proof is provided. Accordingly, we build the system model. The positions of UE1 and UE2 are, respectively, given by ${\left[q_{x},q_{y}\right]}^{T}$ and ${\left[p_{x},p_{y}\right]}^{T}$. As the system is operating in the mmWave band, near-field effects are neglected. The position of RISs, assumed to be points, are represented by ${\left[r1_{x},r1_{y}\right]}^{T}$,${\left[r2_{x},r2_{y}\right]}^{T}$. The rotation angles of the antenna arrays at UE1 and UE2 are denoted by $\gamma_{q},\gamma_{p}\in \left[0,\pi \right]$, respectively. The objective is to locate the positions of UE1 and UE2, given that the positions of RISs are fixed and known.

Consider the system where the working frequency of the carrier is $f_{c}$ and the total bandwidth is *B*. In this system, each UE can control RISs and act as the transmitter for localization tasks. Thus, we assume that UE2 remains silent while UE1 transmits an Orthogonal Frequency Division Multiplexing (OFDM) signal with *N* subcarriers to UE2. Concurrently, UE2 receives both the direct signal from UE1 and the signal reflected from the RISs. During UE1’s transmission, UE1 remains silent as well.

### 2.1. Channel Model

The CSI matrix for the *n*-th subcarrier of the UE1-RIS1-UE2 route is formulated as
(1)$$H_{\mathrm{RIS},1}\left[n\right]=H_{\mathrm{RR},1}\left[n\right]\Omega_{1}H_{\mathrm{TR},1}\left[n\right],$$
where $H_{\mathrm{TR}}\left[n\right]$ is the CSI matrix of the UE1-RIS1 route, $H_{\mathrm{RR}}\left[n\right]$ is the CSI matrix of the RIS1-UE2 route and n∈{−N/2,⋯,N/2}. $\Omega =diag\left(e^{j\omega_{1}},e^{j\omega_{2}},\cdots ,e^{j\omega_{L}}\right)\in C^{L\times L}$ denote the reflection–coefficient matrix at the RIS, where $\omega_{i}\in \left[0,2\pi \right),i=\left\{1,\cdots ,L\right\}$ are the reflection phase shifts. $\Omega_{1}$ and $\Omega_{2}$ denote the reflection–coefficient matrix at RIS1 and RIS2, respectively. $H_{\mathrm{TR},1}\left[n\right]$ and $H_{\mathrm{RR},1}\left[n\right]$ can be written as
(2)$$H_{\mathrm{TR},1}\left[n\right]=\rho_{11}\alpha_{\mathrm{in}}\left(\phi_{\mathrm{in},1}\right)\alpha_{T}^{H}\left(\alpha_{t1}\right)e^{-j2\pi B\frac{n}{N}\tau_{q,r1}},$$
and
(3)$$H_{\mathrm{RR},1}\left[n\right]=\rho_{12}\alpha_{R}\left(\alpha_{r1}\right)\alpha_{\mathrm{out}}^{H}\left(\phi_{\mathrm{out},1}\right)e^{-j2\pi B\frac{n}{N}\tau_{r1,p}},$$
where $\rho_{11}$ and $\rho_{12}$ are the free-space path loss occurred in the UE1-RIS1 and RIS1-UE2 link. $\tau_{q,r1}$ and $\tau_{r1,p}$ are ToAs of the two links. $\alpha_{T}\left(\alpha_{t1}\right)\in C^{N_{u}\times 1}$ and $\alpha_{R}\left(\alpha_{r1}\right)\in C^{N_{u}\times 1}$ are the antenna array steering vectors and response vectors at UE1 and UE2, respectively. The *i*-th entry of $\alpha_{T}\left(\alpha_{t1}\right)$ and $\alpha_{R}\left(\alpha_{r1}\right)$ are ${\left[\alpha_{T}\left(\alpha_{t1}\right)\right]}_{i}=e^{j2\pi \left(i-1\right)\frac{d}{\lambda }sin\left(\alpha_{t1}\right)}$ and ${\left[\alpha_{R}\left(\alpha_{r}\right)\right]}_{i}=e^{j2\pi \left(i-1\right)\frac{d}{\lambda }sin\left(\alpha_{r1}\right)}$, respectively. $\alpha_{t}$ is the AOD of UE1, and $\alpha_{r}$ is the AOA of the UE2. *d* is the antenna element spacing of UE, and RIS1 is the same. $\lambda =c/f_{c}$, with c denoting the speed of light. $\alpha_{\mathrm{in}}\left(\phi_{\mathrm{in}}\right)$ and $\alpha_{\mathrm{out}}\left(\phi_{\mathrm{out}}\right)$ are the array response vectors of the UE1-RIS1 and RIS1-UE2 links, where $\phi_{\mathrm{in}}$ is the AOA at the UE1-RIS1 link, and $\phi_{\mathrm{out}}$ is the AOD at the RIS1-UE2 link. $\alpha_{\mathrm{in},1}\left(\phi_{\mathrm{in}}\right)$ is represented as ${\left[\alpha_{\mathrm{in}}\left(\phi_{\mathrm{in},1}\right)\right]}_{i}=e^{j2\pi \left(i-1\right)\frac{d}{\lambda }sin\left(\phi_{\mathrm{in},1}\right)}$, where i∈{0,1,⋯,L−1}, and $\alpha_{\mathrm{out}}\left(\phi_{\mathrm{out},1}\right)$ is defined similarly. Similar to (Equation 1), (Equation 2), and (Equation 3), the CSI matrix for the *n*-th subcarrier of the UE1-RIS2-UE2 route is formulated as
(4)$$H_{\mathrm{RIS},2}\left[n\right]=H_{\mathrm{RR},2}\left[n\right]\Omega_{2}H_{\mathrm{TR},2}\left[n\right],$$
where
(5)$$H_{\mathrm{TR},2}\left[n\right]=\rho_{21}\alpha_{\mathrm{in}}\left(\phi_{\mathrm{in},2}\right)\alpha_{T}^{H}\left(\alpha_{t2}\right)e^{-j2\pi B\frac{n}{N}\tau_{q,r2}},$$
and
(6)$$H_{\mathrm{RR},2}\left[n\right]=\rho_{22}\alpha_{R}\left(\alpha_{r2}\right)\alpha_{\mathrm{out}}^{H}\left(\phi_{\mathrm{out},2}\right)e^{-j2\pi B\frac{n}{N}\tau_{r2,p}}.$$

Moreover, UE2 will also send a positioning reference signal(PRS) to UE1 after UE2 performs this. The channel state information matrix of the UE2-RIS1-UE1 route is formulated as
(7)$$G_{\mathrm{RIS},1}\left[n\right]=G_{\mathrm{RR},1}\left[n\right]\Omega_{1}G_{\mathrm{TR},1}\left[n\right],$$
where
(8)$$G_{\mathrm{TR},1}\left[n\right]=\rho_{12}\alpha_{\mathrm{out}}\left(\phi_{\mathrm{out},1}\right)\alpha_{T}^{H}\left(\alpha_{r1}\right)e^{-j2\pi B\frac{n}{N}\tau_{p,r1}},$$
(9)$$G_{\mathrm{RR},1}\left[n\right]=\rho_{11}\alpha_{R}\left(\alpha_{t1}\right)\alpha_{\mathrm{in}}^{H}\left(\phi_{\mathrm{in},1}\right)e^{-j2\pi B\frac{n}{N}\tau_{r1,q}}.$$

$\alpha_{T}^{H}\left(\alpha_{r1}\right)$ and $\alpha_{R}\left(\alpha_{t1}\right)$ are the antenna array steering vectors and response vectors at UE2 and UE1, respectively. They are defined similarly as $\alpha_{T}\left(\alpha_{t1}\right)$ and $\alpha_{R}\left(\alpha_{r1}\right)$. Similarly, the CSI matrix $G_{\mathrm{RIS}2}\left[n\right]$ of the UE2-RIS2-UE1 route is expressed as
(10)$$G_{\mathrm{RIS},2}\left[n\right]=G_{\mathrm{RR},2}\left[n\right]\Omega_{2}G_{\mathrm{TR},2}\left[n\right],$$
where
(11)$$G_{\mathrm{TR},2}\left[n\right]=\rho_{22}\alpha_{\mathrm{out}}\left(\phi_{\mathrm{out},2}\right)\alpha_{T}^{H}\left(\alpha_{r1}\right)e^{-j2\pi B\frac{n}{N}\tau_{p,r1}},$$
(12)$$G_{\mathrm{RR},2}\left[n\right]=\rho_{21}\alpha_{R}\left(\alpha_{t2}\right)\alpha_{\mathrm{in}}^{H}\left(\phi_{\mathrm{in},2}\right)e^{-j2\pi B\frac{n}{N}\tau_{r2,q}}.$$

The CSI matrix of the UE1-UE2 route is formulated as
(13)$$H_{\mathrm{UE}2}\left[n\right]=\rho_{3}\alpha_{R}\left(\theta_{r}\right)\alpha_{T}^{H}\left(\theta_{t}\right)e^{-j2\pi B\frac{n}{N}\tau_{u}}.$$

The CSI matrix of the UE2-UE1 route is formulated as
(14)$$G_{\mathrm{UE}1}\left[n\right]=\rho_{3}\alpha_{T}\left(\theta_{t}\right)\alpha_{T}^{H}\left(\theta_{t}\right)e^{-j2\pi B\frac{n}{N}\tau_{u}}.$$

To obtain the distance of UE1 and UE2, each ULA needs to act as a single antenna (A1 and A2) to transmit signals. The CSI matrix of the A1-UE2 route is formulated as
(15)$$h_{\mathrm{UE}2}\left[n\right]=\rho_{3}\alpha_{R}\left(\theta_{r}\right)e^{-j2\pi B\frac{n}{N}\tau_{u}}.$$

The CSI matrix of the A2-UE1 route is written as
(16)$$g_{\mathrm{UE}1}\left[n\right]=\rho_{3}\alpha_{T}\left(\theta_{t}\right)e^{-j2\pi B\frac{n}{N}\tau_{u}},$$
where $\rho_{3}$ is the path loss in the UE1-UE2 link, and $\tau_{u}$ is the path delay. $\alpha_{T}\left(\theta_{t}\right)$ and $\alpha_{T}\left(\theta_{t}\right)$ are defined in the similar way as $\alpha_{T}\left(\alpha_{t1}\right)$.

### 2.2. Receiver Model

Assume that the transmitted PRS is $\mathrm{Fx}\left[n\right]\in C^{N_{u}\times 1}$ where $F\in C^{N_{u}\times N_{u}}$, $x\left[n\right]\in C^{N_{u}\times 1}$, ${\left||\mathrm{Fx}\left[n\right]|\right|}_{2}=1$. F is transmit BF matrix. We assume that F at UE1 is denoted as $F_{1}$, and F at UE2 is denoted as $F_{2}$. According to (Equation 1)–(Equation 16), the signal $y_{1}\left[n\right]$ reflected by the RISs and the signal $y_{3}\left[n\right]$ directly transmitted from UE1 via a single antenna, both of which are received at UE2, can be expressed as
(17)$$\begin{array}{c} y_{1}\left[n\right]=\sqrt{P}\left(H_{\mathrm{RIS},1}\left[n\right]+H_{\mathrm{RIS},2}\left[n\right]+H_{\mathrm{UE}2}\left[n\right]\right)F_{1}x\left[n\right]+n\left[n\right], \end{array}$$
(18)$$\begin{array}{c} y_{3}\left[n\right]=\sqrt{P}h_{\mathrm{UE}2}\left[n\right]+n\left[n\right]. \end{array}$$

The signal $y_{2}\left[n\right],y_{4}\left[n\right]\in C^{N_{u}\times 1}$ received at UE1 can be similarly expressed as
(19)$$\begin{array}{c} y_{2}\left[n\right]=\sqrt{P}\left(G_{\mathrm{RIS},1}\left[n\right]+G_{\mathrm{RIS},2}\left[n\right]+H_{\mathrm{UE}1}\left[n\right]\right)F_{2}x\left[n\right]+n\left[n\right], \end{array}$$
(20)$$\begin{array}{c} y_{4}\left[n\right]=\sqrt{P}g_{\mathrm{UE}1}\left[n\right]+n\left[n\right], \end{array}$$
where $n\left[n\right]\in C^{N_{u}\times 1}$ is the additive white Gaussian noise drawn from $\mathrm{CN}(0,2\sigma^{2}I)$, and I is the identity matrix. *P* is the transmit power of the PRS.

### 2.3. Geometry Relationship

According to the geometric description of the model, the relationship between channel parameters and geometric information can be expressed as
(21)$$\begin{array}{cc} \tau_{\mathrm{ri}} & =\tau_{q,\mathrm{ri}}+\tau_{\mathrm{ri},p}=\frac{d_{\mathrm{ri},q}}{c}+\frac{d_{\mathrm{ri},p}}{c}, \end{array}$$
(22)$$\begin{array}{cc} \tau_{u} & =\frac{d_{q,p}}{c}, \end{array}$$
(23)$$\begin{array}{cc} \theta_{t} & =\pi /2+\arctan\left(\left(p_{x}-q_{x}\right)/\left(p_{y}-q_{y}\right)\right)+\gamma_{q}, \end{array}$$
(24)$$\begin{array}{cc} \theta_{r} & =-\pi /2+\arctan\left(\left(p_{x}-q_{x}\right)/\left(p_{y}-q_{y}\right)\right)+\gamma_{p}, \end{array}$$
(25)$$\begin{array}{cc} \alpha_{\mathrm{ti}} & =\arctan\left(\left({ri}_{y}-q_{y}\right)/\left(q_{x}-{ri}_{x}\right)\right)+\gamma_{q}, \end{array}$$
(26)$$\begin{array}{cc} \alpha_{\mathrm{ri}} & =-\arctan\left(\left({ri}_{y}-q_{y}\right)/\left(q_{x}-{ri}_{x}\right)\right)+\gamma_{p}, \end{array}$$
(27)$$\begin{array}{cc} \phi_{\mathrm{in},i} & =-\arctan(\left({ri}_{y}-q_{y}\right)/\left(q_{x}-ri_{x}\right)), \end{array}$$
(28)$$\begin{array}{cc} \phi_{\mathrm{out},i} & =\arctan(\left(p_{y}-{ri}_{y}\right)/\left(p_{x}-ri_{x}\right)), \end{array}$$
(29)$$\begin{array}{cc} \rho_{i} & =\rho_{i1}\rho_{i2}={\left(d_{\mathrm{ri},q}\right)}^{-\mu /2}{\left(d_{\mathrm{ri},p}\right)}^{-\mu /2}, \end{array}$$
(30)$$\begin{array}{cc} \rho_{3} & ={\left(d_{q,p}\right)}^{-\mu /2}, \end{array}$$
(31)$$\begin{array}{cc} d_{\mathrm{ri},q} & =\sqrt{{\left(q_{x}-ri_{x}\right)}^{2}+{\left(q_{y}-ri_{y}\right)}^{2}}, \end{array}$$
(32)$$\begin{array}{cc} d_{\mathrm{ri},p} & =\sqrt{{\left(p_{x}-ri_{x}\right)}^{2}+{\left(p_{y}-ri_{y}\right)}^{2}}, \end{array}$$
(33)$$\begin{array}{cc} d_{p,q} & =\sqrt{{\left(q_{x}-p_{x}\right)}^{2}+{\left(q_{y}-p_{y}\right)}^{2}}, \end{array}$$
where i∈1,2, μ is the path loss exponent, and $d_{r,q}$, $d_{r,p}$, and $d_{p,q}$ are the distances between the RISs and UE1, RISs and UE2, and UE1 and UE2, respectively.

## 3. Localization and Its Performance Metric

Based on the system model established in Section 2, we can estimate the parameters required for preliminary localization using the methods mentioned in the introduction. Subsequently, precision localization is achieved by adjusting the BF direction based on the preliminary localization results. We employ PEB as the evaluation metric for localization accuracy in this scheme. In Section 4, we will utilize this metric to formulate an objective function and propose a novel joint BF algorithm to optimize this objective function.

### 3.1. Preliminary Localization

Before optimizing the BF, we first roughly estimate the required parameters for localization. We employ the RSSI method for distance estimation and the ESPRIT method for receiving angle estimation. $y_{3}\left[n\right]$ can also be written as:(34)$$\begin{array}{c} y_{3}\left[n,l\right]=\sqrt{P}\rho_{3}e^{j2\pi l\frac{d}{\lambda }sin\left(\theta_{r}\right)}e^{-j2\pi B\frac{n}{N}\tau_{u}}+n\left[n,l\right]. \end{array}$$

Therefore,
(35)$$\begin{array}{c} \hat{\rho_{3}}=\frac{\frac{1}{N+1}\sum_{n=-N/2}^{N/2}\frac{1}{N_{u}}\sum_{l=0}^{N_{u}-1}\left|y_{3}\left[n,l\right]\right|}{\sqrt{P}}. \end{array}$$

We set $Y_{1}=\left[y_{1}\left[-N/2\right],\cdots ,y_{1}\left[N/2\right]\right]$ and calculate its covariance $R=\frac{1}{N+1}Y_{1}Y_{1}^{H}$. Then, $\hat{\theta_{r}}$, ${\hat{\alpha }}_{r1}$, and ${\hat{\alpha }}_{r2}$ can be calculated by R with the aid of ESPRIT algorithm [2]. ${\hat{\theta }}_{t}$, ${\hat{\alpha }}_{t1}$, and ${\hat{\alpha }}_{t2}$ are calculated similarly. Based on the angles and distance, along with the known coordinates of the RISs, we can solve the equations in geometry relationship in Section 2. Thus, we depict the topological spatial location information between devices like Figure 1 and obtain the coordinates of the UEs.

### 3.2. Performance Metric for Localization

PEB is used to measure the localization scheme in a good number of studies. In this section, we derive PEB as a performance metric. As in [10,14], we use a two-stage method to obtain the FIM of the positions of UE1 and UE2. Firstly, we obtain the FIM of the channel parameters η. Then, the FIM for the UEs’ positions can be derived by computing a transformation matrix T that links the two FIM matrices. Finally, we use the FIM of the UEs’ positions to derive the Cramér–Rao Lower bound (CRLB) and position error bound for estimating the UE positions. According to the system model described above, all channel parameters η can be stacked as
(36)$$\eta ={\left[\rho_{3},\theta_{r},\alpha_{r1},\alpha_{r2},\theta_{t},\alpha_{t1},\alpha_{t2}\right]}^{T},$$

We denote the unbiased estimator of η as $\hat{\eta }$. And firstly, we combine all the received signals into a new vector y[n] defined as
(37)$$y\left[n\right]={\left[y_{1}^{T}\left[n\right],y_{2}^{T}\left[n\right],y_{3}^{T}\left[n\right],y_{4}^{T}\left[n\right]\right]}^{T},$$
where $y\left[n\right]∼\mathrm{CN}(\mu \left[n\right],2\sigma^{2}I)$.
(38)$$\mu \left[n\right]=[\mu_{1}^{T}\left[n\right],\mu_{2}^{T}\left[n\right],\mu_{3}^{T}\left[n\right],\mu_{4}^{T}\left[n\right]]$$
where $\mu_{1}\left[n\right]=\sqrt{P}\left(H_{\mathrm{RIS}1}\left[n\right]+H_{\mathrm{RIS}2}\left[n\right]\right)F_{1}x\left[n\right]$, $\mu_{2}\left[n\right]=\sqrt{P}\left(G_{\mathrm{RIS}1}\left[n\right]+G_{\mathrm{RIS}2}\left[n\right]\right)F_{2}x\left[n\right]$, $\mu_{3}\left[n\right]=\sqrt{P}H_{\mathrm{UE}2}\left[n\right]F_{1}x\left[n\right]$, $\mu_{4}\left[n\right]=\sqrt{P}G_{\mathrm{UE}1}\left[n\right]F_{2}x\left[n\right].$

Based on the Cramér–Rao theorem [19,20], the mean squared error of $\hat{\eta }$ is bounded as
(39)$$E\left[\left(\hat{\eta }-\eta \right){\left(\hat{\eta }-\eta \right)}^{H}\right]\geq J_{\eta }^{-1},$$
where $J_{\eta }\in C^{12\times 12}$ is the FIM of η, and ${\left[J_{\eta }^{-1}\right]}_{m,m}$ is the CRLB for the *m*-th parameter estimate. The (m,n)-th entry of $J_{\eta }$ is defined as
(40)$${\left[J_{\eta }\left[n\right]\right]}_{m,n}=E\left[-\frac{\partial^{2}\ln p\left(y\left[n\right];\eta \right)}{\partial \eta_{m}\partial \eta_{n}}\right],$$
where p(y[n];η) is the likelihood function of the random vector y[n] conditioned on η and $\eta_{m}$ is the *m*-th entry of η. Since y[n] follows a symmetric complex normal distribution, the (m,n)-th entry for the FIM can be defined as
(41)$${\left[J_{\eta }\left[n\right]\right]}_{m,n}=\frac{P}{\sigma^{2}}ℜ\left\{\frac{\partial \mu^{H}\left[n\right]}{\partial \eta_{m}}\frac{\partial \mu [n]}{\partial \eta_{n}}\right\},$$

The detailed calculation of μ[n] can be found in Appendix A.

Since our aim is to recover the UEs’ positions $P={\left[q_{x},q_{y},p_{x},p_{y}\right]}^{T}$, we can obtain the FIM $J_{P}$ associated with P by properly transforming $J_{\eta }$, which is shown as follows:(42)$$J_{P}\left[n\right]=TJ_{\eta }\left[n\right]T^{T},$$
where the transformation matrix $T\in C^{4\times 12}$ is defined as:(43)$$T=\frac{\partial \eta^{T}}{\partial P},$$
and the expression of ${\left[T\right]}_{i}$ can be computed according to (Equation 21)–(Equation 33) and found in Appendix B.

Finally, we can obtain the entries of $J_{P}$ as follows,
(44)$${\left[J_{P}\right]}_{m,n}=\sum_{n=-(N-1)/2}^{(N-1)/2}{\left[J_{P}\left[n\right]\right]}_{m,n}=\sum_{n=-(N-1)/2}^{(N-1)/2}{\left[TJ_{\eta }\left[n\right]T^{T}\right]}_{m,n},$$
where m,n∈{0,1,⋯,4}. Therefore, the CRLB for estimating the UEs’ positions is the trace of the inverse matrix of $J_{P}$: $\mathrm{CRLB}=\mathrm{tr}(J_{P}^{-1})$. The PEB can be defined as
(45)$$\mathrm{PEB}=\sqrt{\mathrm{CRLB}}=\sqrt{\mathrm{tr}(J_{P}^{-1})}.$$

## 4. Optimization Objective and Solutions

According to the previous section, we know that the smaller PEB represents the better performance of the positioning system. Thus, in this section, our aim is to minimize the PEB. From Equations (Equation 44) and (Equation 45), it can be seen that PEB is mainly related to $J_{\eta }$ and T. During localization, the channel model and signal model are fixed, and T is a constant matrix. Therefore, PEB depends on the BF matrix and the reflection phase shifts at the RIS. Hence, we can minimize PEB by altering F and Ω.

Considering practical constraints, assume that the phase shift of each reflection element can only take discrete values. Let $Q_{r}$ represent the number of control bits for each element, and suppose that the values of the discrete phase shifts are uniformly distributed in the range (0,2π]. In Ω, $\omega_{i}\in F$, where $F=\{0,\frac{2\pi }{Q_{r}},\cdots ,\frac{2\pi (Q_{r}-1)}{Q_{r}}\}$. In addition, assume that the transmit BF matrix $F=[f_{1},\cdots ,f_{N_{u}}]$ is constrained to take values from a codebook, where $f_{k}={\left[e^{jv_{1}},\cdots ,e^{jv_{N_{u}}}\right]}^{T}$. We have $\mathrm{Fx}\left[n\right]=1/N_{u}{\left[e^{jv_{1}},\cdots ,e^{jv_{N_{u}}}\right]}^{T}$, where $v_{j}\in G$, $G=\{0,\frac{2\pi }{Q_{f}},\cdots ,\frac{2\pi (Q_{f}-1)}{Q_{f}}\}$. Therefore, we can define an optimization objective function as
(46)$$\begin{array}{cc} \; & \min_{w_{i}\forall i\in \left\{1,\ldots ,L\;\right\},v_{j}\forall j\in \left\{1,\ldots ,N_{u}\right\}}\sqrt{\mathrm{tr}(J_{P}^{-1})}\\ s.t. & \;\omega_{i}\in F,v_{j}\in G,\;\forall i=1,\ldots ,L,\forall j=1,\ldots ,N_{u}. \end{array}$$

Since this objective function is non-convex, the block coordinate descent (BCD) algorithm can help us, which can be found in [14,21]. Hence, we propose a BCD-based joint alternative optimization as illustrated in Algorithm 1 and we call this optimization the joint BF design. In each iteration, we optimize $\omega_{i}$ while keeping $\omega_{k}$ fixed, where $k\in \{1,\cdots i-1,i+1,\cdots ,L^{2}\}$, and optimize $v_{j}$ while keeping $v_{l}$ fixed, where $l\in \{1,\cdots ,j-1,j+1,\cdots \;N_{u}\}$. As a result, in $\Omega_{1}$ or $\Omega_{2}$, we have
(47)$$\hat{\omega_{i}}=\arg\min_{w_{i}\in F}\sqrt{\mathrm{tr}(J_{P}^{-1})},$$
where i∈{1,…,L}, and in $F_{1}$ or $F_{2}$, we have
(48)$$\hat{v_{j}}=\arg\min_{v_{j}\in G}\sqrt{\mathrm{tr}(J_{P}^{-1})},$$
where $j\in \{1,\ldots ,N_{u}\}$. In order to approach the optimal solution as closely as possible, we alternate between optimizing Ω while fixing F and optimizing F while fixing Ω. We calculate minimum PEB after optimizing Ω and F.

Finally, we present a concise complexity analysis of the proposed algorithm, which is pertinent given that many BCD-based algorithms undergo similar analyses. As delineated in Algorithm 1, during the *i*-th cycle of the *t*-th iteration, step 6 incurs a complexity of approximately $O(N_{u}^{3}L^{4}N)$ operations. This complexity primarily stems from the multiplication and inversion of matrices. Notably, step 9 mirrors the operations of step 6. Consequently, the total complexity of the proposed algorithm is expressed as $O(T\left(LQ_{r}+N_{u}Q_{f}\right)\left(N_{u}^{3}L^{4}N\right))$, with *T*, typically less than $10^{1}$, representing the number of iterations for Algorithm 1.
**Algorithm 1** The Proposed Joint Beamforming Design1:$\mathrm{Objective}\;\mathrm{Function}:\;\mathrm{PEB}=\sqrt{\mathrm{tr}(J_{P}^{-1})}$2:$\mathrm{Initialize}\;\eta ,\Omega_{0},F_{0},\epsilon ,{\mathrm{PEB}}_{0}\;\mathrm{and}\;\mathrm{set}\;t,i,j:=0.$3:**while** 
$\;\Delta \mathrm{PEB}=|{\mathrm{PEB}}_{t}-{\mathrm{PEB}}_{t-1}|>\epsilon$ **do**4:    t:=t+15:    **for** i=1toL **do**6:        $\;\mathrm{Update}\;\hat{\omega_{i}}\in F\;s.t.\;\hat{\omega_{i}}=\arg\min\sqrt{\mathrm{tr}(J_{P}^{-1})}.$7:    **end for**8:    **for** $j=1\;\mathrm{to}\;N_{u}$ **do**9:        $\;\mathrm{Update}\;\hat{v_{j}}\in G\;s.t.\;\hat{v_{j}}=\arg\min\sqrt{\mathrm{tr}(J_{P}^{-1})}.$10:    **end for**11:    $\mathrm{Calculate}\;{\mathrm{PEB}}_{t}\;\mathrm{and}\;\Delta \mathrm{PEB}$12:**end while**13:$\mathrm{Calculate}\;\mathrm{PEB}\;\mathrm{with}\;\Omega_{1}^{*},\Omega_{2}^{*}\;\mathrm{and}\;F_{1}^{*},F_{2}^{*}.$

## 5. Simulation Analysis

In this section, we analyze PEB to demonstrate the performance of the proposed system. Firstly, considering far field constraints, the position of the RISs are set at coordinates [0,0] m and [0,20] m, UE1 is at [20,−10] m, and UE2 is at [20,30] m. The signal-to-noise ratio (SNR) is defined as $\frac{P}{\sigma^{2}}$. If not specified, the default parameter settings are listed in Table 1. These parameter settings have taken into account the avoidance of near-field effects.

### 5.1. Impact of the Joint BF Design

Firstly, we investigate the impact of the BF design on the transmitter and RIS on PEB. As shown in Figure 2, we compare four different cases, including (1) random F and random Ω; (2) random F and optimized Ω; (3) optimized F and random Ω; and (4) the joint optimized BF design based on the iterative solution of (Equation 46) by Algorithm 1.

The PEB performance of these four designs is depicted across various SNR levels in Figure 2. The results indicate that optimized Ω or F yields superior PEB performance compared to random Ω or F, with PEB decreasing as SNR increases. This phenomenon arises from the ability to align transmitting beams, thereby concentrating them to achieve higher transmission gains. Similarly, without optimizing the phase design of the RIS, random reflection fails to attain higher acceptance gain for the receiving antenna. Of significant note, the joint BF design at the transmitter–RIS, based on Equation (Equation 44), achieves optimal PEB for the localization scheme, enabling centimeter-level localization accuracy under the given conditions.

### 5.2. Impact of $Q_{r}$ and $Q_{f}$

Next, we consider the impact of different phase quantization resolutions on the PEB generated by the joint BF design. As shown in Figure 3, it is evident that PEB decreases as $Q_{r}$ and $Q_{f}$ increase. However, it can also be observed that PEB growth saturates when $Q_{r}\geq 6$ or $Q_{f}\geq 6$.

### 5.3. Impact of L, $N_{u}$, and N

Next, we consider the impact of subcarrier count design and the size design of RIS on the localization performance. As shown in Figure 4 and Figure 5, it is obvious that with the increase of *L*, $N_{u}$, and *N*, a significant reduction in PEB can be observed.

This is intuitive, as a larger RIS represents more RIS reflecting elements, thereby providing more reflective links to deliver the required localization information. Having longer antennas on the UEs implies the capability to transmit and receive more link information, naturally aiding in improving localization accuracy. Similarly, a larger *N* implies more subcarriers, which also brings more links capable of providing measurement information. The results further indicate that by selecting an appropriately sized RIS, achieving centimeter-level localization accuracy can be easily realized with a large number of reflecting elements.

### 5.4. Impact of the Distance between UEs

Furthermore, the distance between UE1 and UE2 should also be considered. Constrained by the signal attenuation in free space, PEB gradually increases as the distance between devices grows larger in Figure 6. It is also noted that increasing power can partially compensate for this loss of accuracy.

### 5.5. Comparison

Finally, we compare the performance gap between the proposed scheme and the scheme (Comparison) that BS locates UEs with the assistance of RISs as in Figure 7. A similar model can be seen in [11] for the localization scheme used for comparison. It is essential to note that the comparative scheme under consideration lacks D2D links. Furthermore, in practical scenarios, the transmission power of the BS, at $P_{t}=10\;\mathrm{dBm}$, is significantly higher than that of the UEs. Typically, the distance $d^{\prime }$ between the BS and the UEs should far exceed the distance between the UEs themselves. Accordingly, we have set $d^{\prime }=5\times d_{p,q}$. The BS is positioned at coordinates [20, −170] meters and is equipped with a single antenna, hindered by obstacles. Localization can only be achieved through reflection with the assistance of RISs. Other parameters were set to be as consistent as possible.

As can be seen from Figure 8, the proposed scheme clearly outperforms the traditional RIS-assisted localization and lower power can yield improved performance. Additionally, replacing RISs with a larger size or adding more RISs can bridge the performance gap between the schemes. This demonstrates the feasibility and advantages of cooperative localization schemes. However, it should be noted that our proposed scheme requires collaborative transmission of signals among devices, whereas the comparative scheme only requires a single transmission. As a result, our scheme incurs twice the complexity. Moreover, this D2D localization scheme distributes the localization burden to UEs rather than BS, reducing the power pressure on BS. Even as a performance supplement for conventional localization schemes, it remains a viable option, as can be seen in Table 2.

## 6. Conclusions

In this paper, we proposed a system for D2D cooperative localization in mmWave MIMO systems using RISs as anchor points, without relying on BS. UEs transmit PRS to each other through RISs to achieve localization. A system model is then established based on this scheme. The PEB, which evaluates the coordinate error of the two UEs’ positions, is derived, and a joint transmitter–RIS BF design algorithm based on alternating optimization is proposed to minimize the PEB. The simulation results demonstrate that the proposed algorithm significantly improves the localization performance and the proposed scheme can achieve centimeter-level localization accuracy.

In conclusion, our proposed cooperative localization scheme leveraging RISs presents a promising solution for enhancing localization accuracy and flexibility in mmWave MIMO systems. Future research directions may include further optimization algorithms and practical implementations to validate the effectiveness of our proposed scheme in real-world scenarios.

## Figures and Tables

**Figure 1 sensors-24-03694-f001:**
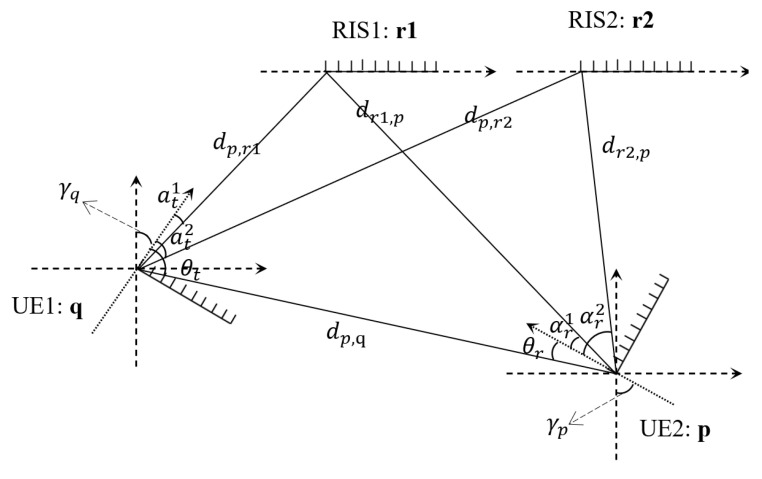
Localization system with one *L*-antenna RIS and two $N_{u}$-antenna UEs.

**Figure 2 sensors-24-03694-f002:**
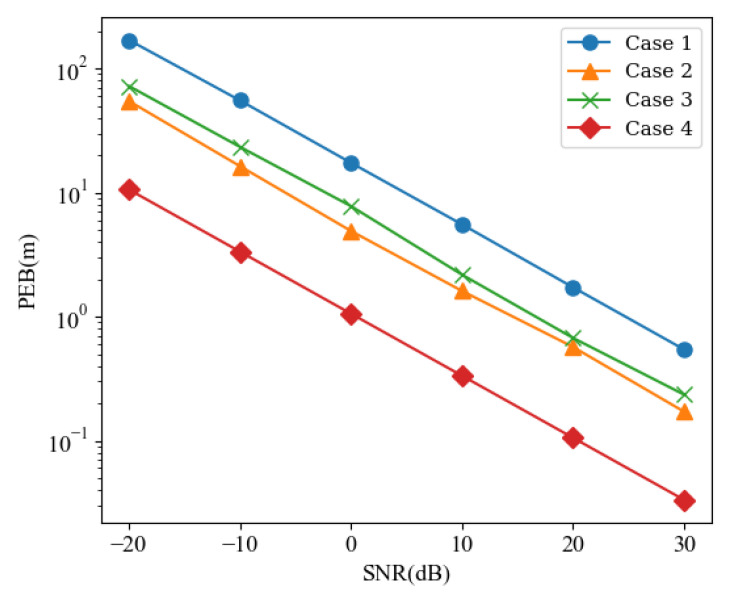
PEB versus SNRs for different BF design.

**Figure 3 sensors-24-03694-f003:**
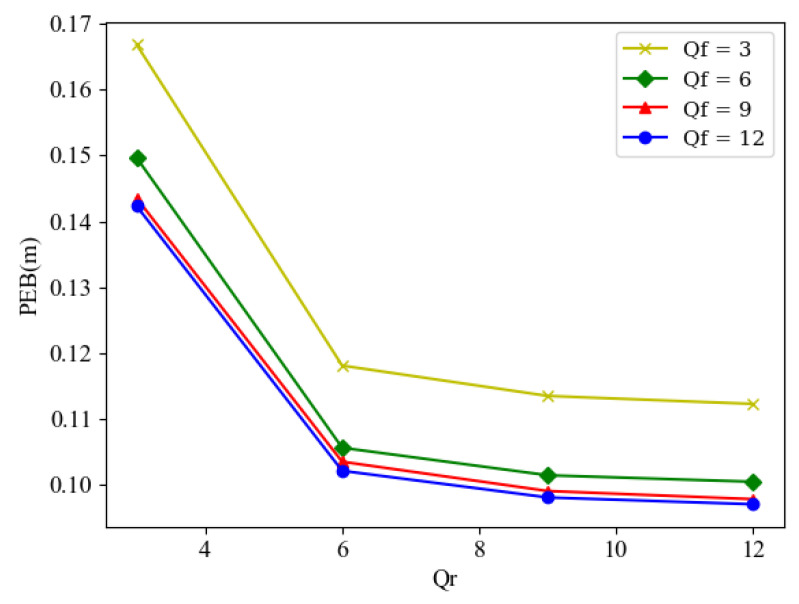
PEB versus different $Q_{r}$ for different $Q_{f}$.

**Figure 4 sensors-24-03694-f004:**
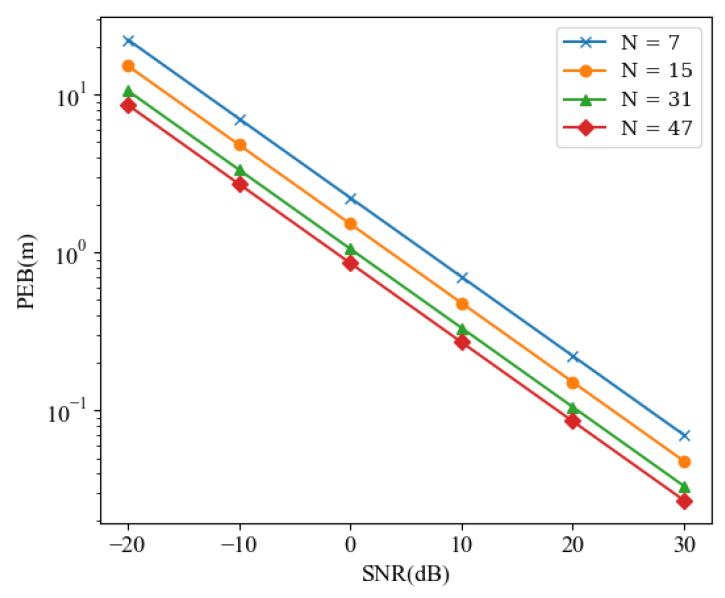
PEB versus different SNRs for different *N*.

**Figure 5 sensors-24-03694-f005:**
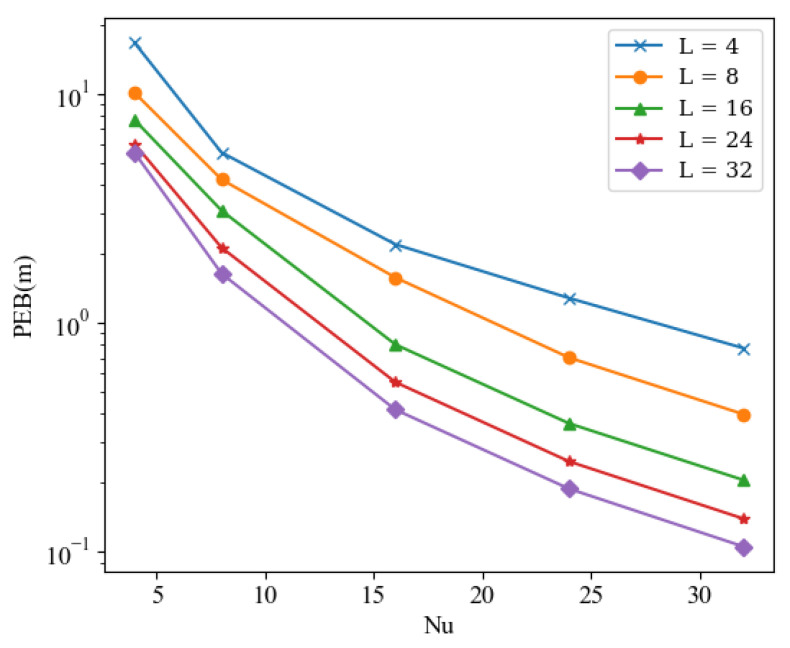
PEB versus different $N_{u}$ for different *L*.

**Figure 6 sensors-24-03694-f006:**
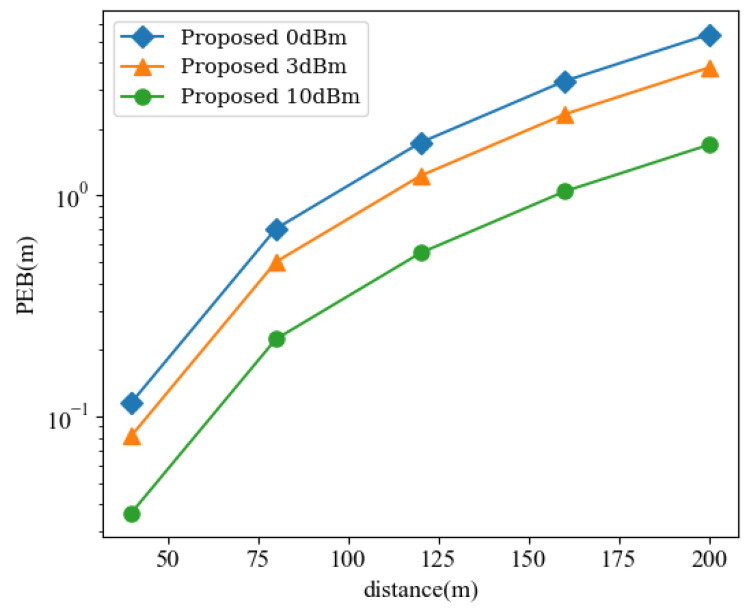
PEB versus different $d_{q,p}$ between UE1 and UE2 for different *P*.

**Figure 7 sensors-24-03694-f007:**
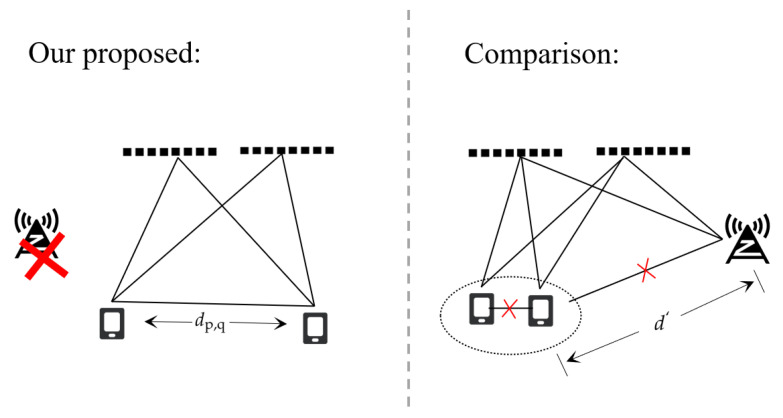
Scenario difference between ours and the comparison.

**Figure 8 sensors-24-03694-f008:**
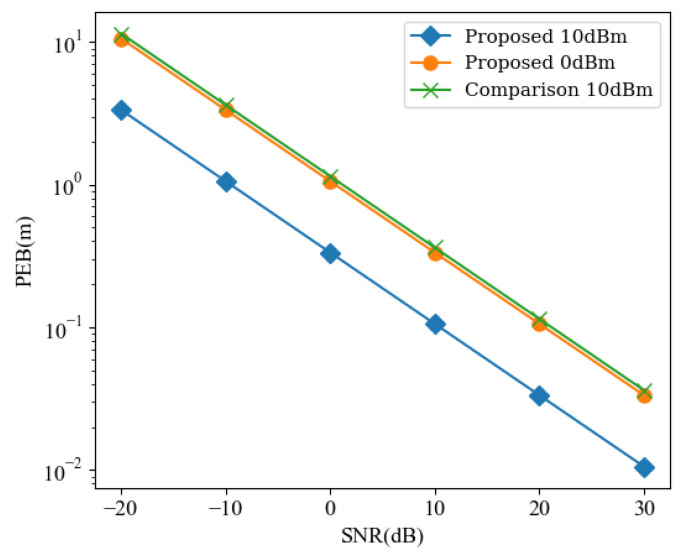
PEB versus different SNRs for different scheme.

**Table 1 sensors-24-03694-t001:** Environmental parameters.

Parameters	Values	Description
*L*	32	The number of reflecting antenna elements for each RIS
$Q_{f}$	6	The number of control bits for each UE
$Q_{r}$	6	The number of control bits for each RIS
$N_{u}$	32	The number of ULA antennas
*P*	0 dBm	The transmit power of PRS
$\gamma_{q}$	−π/9	The rotation angle of UE1
$\gamma_{p}$	5π/12	The rotation angle of UE2
*N*	31	The number of subcarriers
ϵ	$10^{-6}$	The accuracy rating of Algorithm 1
*d*	0.005 m	The spacing between antenna arrays
$f_{c}$	30 GHz	The center frequency of PRS
*B*	100 MHz	The subcarriers bandwidth
*c*	$3\times 10^{8}\;m/s$	The velocity of light
μ	2.08 [10]	The path loss exponent
$\sigma^{2}$	−80 dBm [7]	The free space noise power

**Table 2 sensors-24-03694-t002:** Difference between our proposed and comparison.

Points	Our Proposed	Comparison
Distance between transmitter and receiver	40 m	200 m
Transmission power	0 dBm	10 dBm
Transmission path	not blocked and collaborative	blocked
Localization performance	better	good
Time complexity	double the comparison	normal

## Data Availability

Data are contained within the article.

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
