# Peer review of "Localization Performance Analysis and Algorithm Design of Reconfigurable Intelligent Surface-Assisted D2D Systems"

_sensors, 2024, doi:10.3390/s24113694_

Round 1
Reviewer 1 Report
Comments and Suggestions for Authors
The manuscript introduces a cooperative localization system for mmWave MIMO systems, utilizing RIS as anchors. The authors' innovative joint beamforming design greatly enhances positioning accuracy, yielding valuable results. This work substantially progresses the field of wireless localization, adeptly addressing challenging non-line-of-sight conditions with a meticulous analysis and a robust solution.
To further elevate the manuscript's quality and impact:
-
The current literature review does not include references from 2023 onwards. It would be beneficial to incorporate the latest research in wireless communication and localization to ensure the manuscript's relevance and comprehensiveness.
- Ensure all mathematical equations are properly sourced, it is equally important for academic integrity and context. I recommend the authors to cite all mathematical formulations to enhance the paper's relevance and credibility.
- The comparative analysis in "section 5.5" is beneficial but could be enhanced with detailed performance metrics. Including data on positioning error bounds, computational complexity, and energy efficiency would offer a more thorough evaluation. A comparative table or chart could distill the key differences, emphasizing the proposed system's advantages and practicality.
Reviewer 2 Report
Comments and Suggestions for Authors
This paper proposed a cooperative localization system assisted by RISs, where the D2D communication between UEs are considered. The reviewer has the following comments:
1. The title of this manuscript is not appropriate, there is no performance analysis in this work.
2. In the introduction, the statement “the studies above are all based on the localization method with the BS as the anchor point” is not accurate. In [10] and [11], the RIS is also regarded as anchor points.
3. What does y_3 in (18) stands for?
4. Although alternating optimization is employed to optimize the transmit beamforming and RIS coefficients, this does not mean that the optimal solution can be obtained. Some of the statement should be modified, such as “In order to converge to the optimal solution”. Besides, as previous works have proposed similar ideas for localization before, what are the new technical contributions? The previous works should be acknowledged in this part.
5. In the simulations, such as Fig. 2, the SNR is set too high, which is not practical. Besides, the comparison scheme is not clearly explained. Is there user cooperation in this comparison scheme?
6. The presentation of this work should be improved, there are many typos in the manuscript.
Comments on the Quality of English LanguageThe presentation of this work should be improved, there are many typos in the manuscript.
Reviewer 3 Report
Comments and Suggestions for Authors
Add some numerical results at the end of the abstract.
There is no synchronization while abbreviating the terms in the paper, i.e., Multiple Signal Classification (MUSIC) and received signal strength indicator (RSSI). Please review the manuscript to correct it.
Please provide the reference for each parameter value you have taken in the simulation section.
The assertion that "Two or more RIS units are necessary because we need a sufficient number of anchor points to construct the spatial topology of the devices" relies on geometric intuition without formal proof. However, determining the optimal number of RIS units is crucial for practical deployment and performance optimization of the proposed system. I recommend that the authors include a formal analysis or a simulation-based study to compute the optimal number of RIS units required. This analysis should consider various factors such as the coverage area, the number of UEs, the desired localization accuracy, and the impact of environmental conditions.
The paper would benefit from including a graph that depicts the Root Mean Square Error (RMSE) versus transmit power under different numbers of reflecting elements. This graph is crucial as it visually demonstrates how the localization accuracy varies with changes in power and the number of reflecting elements.
The paper lacks a complexity analysis of the proposed algorithm.
Comments on the Quality of English LanguageThe paper is generally well-written, but there are a few minor grammatical errors and awkward phrasings that could be improved for better readability. For example, in the abstract, the phrase "considering device-to-device (D2D) communication" would be clearer as "which considers device-to-device (D2D) communication." Additionally, "We illustrate our objective function and idetail the algorithm steps" should be corrected to "We illustrate our objective function and detail the algorithm steps." Careful proofreading to catch such small errors would enhance the overall quality of the paper.
Round 2
Reviewer 2 Report
Comments and Suggestions for Authors
No further comments.
Comments on the Quality of English LanguageNone
Reviewer 3 Report
Comments and Suggestions for Authors
I have no further comments.
All my comments are incorporated in the revised version.